# From Substitute to Supported Decision Making: Practitioner, Community and Service-User Perspectives on Privileging Will and Preferences in Mental Health Care

**DOI:** 10.3390/ijerph19106002

**Published:** 2022-05-15

**Authors:** Sarah Gordon, Tracey Gardiner, Kris Gledhill, Armon Tamatea, Giles Newton-Howes

**Affiliations:** 1Department of Psychological Medicine, University of Otago Wellington, P.O. Box 7343, Wellington 6242, New Zealand; tracey.gardiner@otago.ac.nz (T.G.); giles.newton-howes@otago.ac.nz (G.N.-H.); 2Law School, Faculty of Business, Economics and Law, Auckland University of Technology, Private Bag 92006, Auckland 1142, New Zealand; kris.gledhill@aut.ac.nz; 3Te Kura Whatu Oho Mauri, School of Psychology, University of Waikato, Private Bag 3105, Hamilton 3240, New Zealand; armon.tamatea@waikato.ac.nz

**Keywords:** human rights, supported decision making, Convention on the Rights of Persons with Disabilities, mental distress, indigenous peoples

## Abstract

Compliance with the Convention on the Rights of Persons with Disabilities (CRPD) requires substitute decision making being abolished and replaced with supported decision making. The current exploratory study involved a series of hui (meetings) with subject matter experts across the spectrum of the mental health care system to identify interventions facilitative of supported decision making; and the prioritisation of those in accordance with their own perspectives. A mixed-methods approach was used to categorise, describe and rank the data. Categories of intervention identified included proactive pre-event planning/post-event debriefing, enabling options and choices, information provision, facilitating conditions and support to make a decision, and education. The category of facilitating conditions and support to make a decision was prioritised by the majority of stakeholders; however, people from Māori, Pasifika, and LGBTQIA+ perspectives, who disproportionally experience inequities and discrimination, prioritised the categories of proactive post-event debriefing/pre-event planning and/or information provision. Similar attributes across categories of intervention detailed the importance of easily and variably accessible options and choices and how these could best be supported in terms of people, place, time, material resources, regular reviews and reflection. Implications of these findings, particularly in terms of the operationalisation of supported decision making in practice, are discussed.

## 1. Introduction

The Convention on the Rights of Persons with Disabilities (CRPD) is an international human rights instrument that clarifies the application of existing human rights to persons with disabilities, a group whose membership is not fully defined but does include “*those who have long-term physical, mental, intellectual or sensory impairments which in interaction with various barriers may hinder their full and effective participation in society on an equal basis with others.*” [1].

As of this writing, 182 states or regional organisations had ratified the CRPD (note: some declarations and reservations were made upon ratification, formal confirmation or accession), thereby becoming obliged under international law to implement it [2]. This requires “appropriate legislative, administrative and other measures”, including modifying discriminatory “customs and practices” and supporting relevant research and training [3]. Monitoring implementation involves both a domestic process and regular reports to the Committee on the Rights of Persons with Disabilities (CRPD Committee) [4]. 

Psychiatry has had a long history of practice being underpinned by compulsion, authorised by legal frameworks, whereby on the basis of status (diagnosis), outcome (negative consequences), or functional (where a person’s decision-making skills are considered to be deficient) judgments, persons lose the legal right to make decisions about treatment. Decisions are instead made by others (clinicians and courts), a process of “substituted decision making”, which rests on the substitute decision maker determining what they believe to be in the “best interests” of the person. 

Reflecting its importance and to preclude any misunderstanding of what it required, the first General Comment produced by the CRPD Committee contained further guidance as to the obligations deriving from Article 12 of the Convention—Equal recognition before the law [5]. The General Comment emphasises that legal capacity—the right to make decisions about oneself—is fundamental, such that an assessment of perceived or actual impaired decision-making skills should result in the provision of the support necessary to enable individuals to make decisions in accordance with their will and preferences: supported decision making. 

In New Zealand, where this present study is situated, the Mental Health Act is now thirty years old. This Act enables a psychiatrist to substitute the decision making of an individual based on an assessment of mental disorder (including an intermittent disorder) and risk. Both so-called limbs are broad in their interpretation. Numerous international and national critiques of the Act itself and its use resulted in the most recent mental health service inquiry recommending the immediate repeal and replacement of the Act, the process of which has now commenced [6]. Details associated with the process [7] as well as guidelines for short-term improvements to the way the Mental Health Act functions now [8] have both signalled that supported decision making will be a key feature of the reformed legislation. 

However, “…legislative change on its own will not drive systemic change… Legislative change also needs to be supported by clear guidance and clinical best practice that promotes supported decision-making and provides measures to minimise compulsory or coercive treatment” [6] (p. 194). Treaty bodies monitoring other human rights treaties have indicated that appropriate measures for implementation of human rights obligations may include legislative, judicial, administrative, educational, financial and social measures [9]. In support of this, and based on the CRPD, a decade ago, the World Health Organisation (WHO) Quality Rights global initiative made available a toolkit that provides practical information and tools for assessing and improving quality and human rights standards in mental health and social care facilities [10].

The academic literature to date has largely involved critiques of the CRPD committee’s guidance as to Article 12 obligations, and particularly the potential for adverse outcomes in response, e.g. [11,12,13,14,15,16,17]. There is a dearth of research and literature focused on informing how supported decision making may best be facilitated, although this appears to be improving, with a number of recent published studies, reviews and resources of applicability [18,19,20,21,22]. Based on crisis response practices that research shows—or that clinical and advocacy practice suggests—are anchored in human rights, Statsny et al., propose nine critical elements of a rights-based approach to crisis: communication and dialogue, presence (“being with”), flexible location, safe spaces of respite, continuity, peer involvement, harm reduction, judicious use of medications, and response to basic needs [18]. An expansive literature review identified 10 studies across six countries where an explicitly human rights-based approach was used to improve mental health outcomes [19]. The actual interventions included human rights training; integration of human rights principles into organisational policy, strategy, and action plans; patient involvement forum; development of human rights auditing, indicators and benchmarking tools; person-centred planning; use of human rights-based approaches to clinical decision making and service supervision. A pilot study that applied some such interventions as part of an overarching human rights-based approach to psychiatric care in a region of Sweden concluded that time for reflection and education was the single most important investment in the pilot project [20]. Services and methods that could be used to realise the rights of service users, and particularly the right to supported decision making, have been identified as including peer support, a circle of support, open dialogue, a circle of care, a personal ombudsman, a crisis plan, a crisis card, and a crisis care centre or house [21]. Indeed, many of these feature in the recently published WHO resource that provides real-world examples of various services that promote person-centred and rights-based approaches to mental health systems and services based on the underpinning core principles of respect for legal capacity, non-coercive practices, community inclusion, participation and the recovery approach [22]. These studies, reviews and resources demonstrate that supported decision making can be successfully applied in various situations, including those of an acute nature, and result in better or, at least, no worse outcomes, with potential cost savings, although all do note that considerably more research is needed to evaluate impact [18,19,20,21,22].

Whilst the identification of existing service and practice exemplars that can facilitate supported decision making is most welcome and useful, it could be potentially limiting to focus solely on these. Many would not have been developed specifically with supported decision making in mind, nor in a manner that reflected the priorities of a wide range of stakeholders, particularly in a culturally informed context. The present work was conducted within Aotearoa New Zealand, a country where a Treaty exists that governs relations between Māori—the indigenous people of New Zealand—and all others. The Treaty—Te Tiriti o Waitangi—emphasises the protection of the collective interests of Māori as part of a partnership accord with the Crown and requires that research be proactively inclusive of, and responsive to, Māori issues, needs and aspirations (outcomes for Māori). Ethical imperatives require similar inclusivity and responsivity in relation to people from other perspectives who also face inequities and often experience overlapping and interdependent systems of discrimination or disadvantage (intersectionality). 

The aforementioned point to the following research questions in relation to enacting supported decision making: (1) What are potentially useful interventions? (2) How would preferences for these interventions be reflected across diverse stakeholder groups, and Māori in particular (for the Aotearoa New Zealand context)? In response, we devised a co-produced, exploratory, mixed-methods study that involved a bottom-up process that brought key stakeholders together and enabled the identification and prioritising of interventions facilitative of supported decision making that they considered as having the potential to lead to the greatest improvements in the lives of people who experience mental distress and their loved ones. This study consequently contributes to the developing scholarship in this area and serves to further inform how supported decision making can be operationalised in practice. 

## 2. Materials and Methods

The aim of this study was to elicit and explore useful interventions from stakeholders as well as ascertain the preferences for each class of interventions by stakeholder group. As such, this study adopted a co-produced [23], exploratory, mixed-methods study design [24,25] to capture the ideas from the participants as well as rank order the preferred approaches to compare across groups. 

### 2.1. Co-Production

This study was co-produced by way of all authors, an academic team that reflected service user, clinical, legal and Māori perspectives, fully collaborating in all stages of the research.

### 2.2. Procedure

There are currently 20 District Health Boards (DHBs) in New Zealand, who are responsible for providing or funding the provision of public health services in their district. Health care in New Zealand is both publicly and privately delivered, although almost all mental health provision occurs in the public domain. Six DHBs were engaged to host and support stakeholders of their organisations and within their geographical area to attend and participate in a hui (see below). The DHBs were chosen based on those that expressed interest in supporting the research, with an emphasis on including urban and rural localities.

*Hui* (a gathering of people operating within Māori protocols—greeting, engagement, connection, consultation, and conclusion) was adopted as a mode of connection and data gathering [26]. Hui promotes the primacy of bringing people together to ‘talk’ in an interactive encounter that reduces perceived status amongst participants so that the expression of viewpoints are roundly respected and the context for speaking is *tika* (‘just’). This method of coming together for the purposes of knowledge generation particularly supports the participation of those who tend to be marginalised through conventional spaces and methods of engagement. The principles of *pōwhiri* [27] are premised on the notion of respect and positive relationships between the *tangata whenua* (‘people of the area’, in this case, the hosts or research participants) and *manuhiri* (‘guests from outside of the area’, in this case, the researchers). This culturally salient approach is particularly important in the Aotearoa New Zealand context given that it is recognised that Māori are significantly proportionally over-represented among people who experience mental distress [6] and that researchers have obligations to actively contribute to Māori health advancement. 

For each of the six hui, invitations to attend and participate were disseminated by the host DHBs to key stakeholders, including clinicians (medicine, nursing, and allied health), health care support workers/assistants, consumers, Māori and other cultural advisors, service users, *tangata whaiora* (‘a person seeking wellness’), family, and *whānau* (a broader notion of family, but not exclusively limited to kinship ties). The researchers also extended invitations through their contacts, and networks of stakeholders they were aware of, in each of the regions. 

Stakeholders who expressed interest were then sent an information sheet and all who proceeded to register provided written informed consent to attend and participate at the hui. Prior to the hui participants also provided demographic information and completed an online survey. The ethnicity question asked people to tick all that applied from the six major ethnic groups in New Zealand: Pākehā (European of New Zealand descent), Māori, Pacific peoples, Asian, MELAA (Middle Eastern/Latin American/African), and ‘Other ethnicity’. The survey ascertained the perceived level of key stakeholder knowledge in respect of supported decision making and any interventions considered to be currently in existence that were facilitative of supportive decision making.

Each of the hui were three and a half hours long in order to allow sufficient time for the proposed activity to be undertaken in a manner that supported appropriate process and facilitation of attendance to the aims of this study. Appropriate processes involved the observance of Māori protocols, including *whakatau* (semi-formal welcome), sharing of *kai* (food), and *mihimihi* (an oral introduction used to establish links with other people present in the meeting space and to let people know who one is and where one is from) prior to commencing any subject-focused activity, and *poroporoaki* (conclusion of proceedings and farewell) at the conclusion of each hui.

The subject-focused activity of the hui commenced with a short 30 min briefing on the background to, and the key elements of, the concept of supported decision making; and the importance of supported decision making from service user, legal, clinical, and Māori perspectives.

Participants were then invited to work in smaller groups of 3–6 people to identify three possible interventions facilitative of supported decision making to include what it would involve, how it would be facilitated, and by whom (both individually and organisationally). More specifically, participants were asked to record each intervention and all associated details on an A2 piece of paper. At the conclusion of this activity, each group presented the three possible interventions they had identified to the wider group.

An advanced form of dot voting (“dotmocracy”) was used to facilitate stakeholder prioritisation of the identified possible interventions. This was justified as a simple tool used to prioritise items within a group setting. However, potential limitations are that it does not adequately support the views of those that come from minority perspectives; and the relative prioritisation of various options. Hence, the process also involved people using labels that were colour coded (to reflect 1st, 2nd, and 3rd preferences), and self-identification of the perspectives each stakeholder came from, which were then stuck to the various A2 pieces of paper (containing the interventions that had been identified by each group) to facilitate the ranking of preferred options of interventions. Self-identification was entirely left to participant choice, with free text to make this identification (this then requiring the collapsing of these identifiers into similar groups during analysis). At the conclusion of each hui, all information presented on each of the A2 pieces of paper, including intervention details and the assigned labels reflecting stakeholder perspective and preference, were transcribed verbatim.

### 2.3. Analysis

A mixed-methods approach was used to categorise and describe—by theme—the interventions and their key attributes as had been identified by stakeholders (qualitative) before calculating ordinal rankings of preferences overall and by perspective for comparison across groups (quantitative). 

The information collected from each hui was initially written up in its raw form. After reading the raw data to achieve familiarity, the lead researcher led a basic thematic analysis [28] of this data which involved both between-category and within-category analysis of the data. For the between-category analysis, the first cycle of coding involved each of the individually identified interventions being labelled. The second cycle involved labelled interventions of similarity being grouped together as provisional categories of intervention [between-category analysis]. All members of the research team reviewed and conferred over the provisional categories to attain collective agreement. Each provisional category was then composed to include the similar interventions and the details associated with each of them. The lead researcher then identified similar key attributes within the detail associated with each of the categorised interventions [within-category analysis]. This involved codes being assigned to the material initially. Materials with codes of similarity were then grouped together, as provisional key attributes of each of the categories. All members of the research team reviewed and conferred over these provisionally proposed attributes to attain collective agreement. At every stage, the data in raw form were returned to multiple times to confirm that everything of relevance was included.

As self-identified stakeholder perspectives were not constrained, those of similarity to each other were combined and categorised based on groupings determined by the researchers (e.g., the category of service users included those that self-identified as service user, lived experience, patient, and peer). Rankings of each intervention were assigned a numerical value of three for 1st preferences, two for 2nd preferences, and one for the 3rd preferences of stakeholders in order to calculate ordinal rankings of preferences. Where multiple perspectives were self-identified, the assigned preference value was included in each perspective category of relevance. An overall weighting rank for each category of interventions was determined for participants overall and then by perspective for all those categories that included at least ten stakeholders. This information was then compared and contrasted to identify any points of difference between stakeholder groups from different perspectives. 

### 2.4. Ethics

This study was conducted according to the guidelines of the Declaration of Helsinki and was also informed by Indigenous research ethics [29,30]. Approval was obtained by the Ethics Committee of the University of Otago (protocol code D20/435, 18 December 2020).

## 3. Results

The number of participants that attended each hui ranged from 10 to 17, giving a total of 83 study participants. The demographics of hui participants are presented in Table 1. Stakeholders primarily attended hui and participated in the capacity of service providers, service users, and peer support workers/advisors. Participants identified a range of ethnicities with a significant number coming from a Māori perspective. Most participants were above 35 and had spent lengthy periods of time in the sector. There were considerably more female participants than male.

The results of the pre-hui survey are presented in Table 2. Pre-hui, most participants self-described as having had little to average knowledge about supported decision making generally, the difference between legal capacity and mental capacity, and the key principles underpinning supported decision making. Less was felt to be known about the difference between substitute decision making and supported decision making, and the current status of supported decision making in New Zealand. Participants had the greatest level of knowledge of guidelines that were developed and published in 2020 as an interim measure whilst the current Mental Health Act is undergoing repeal and replacement. The guidance provided encompasses how to think about and apply human rights, recovery approaches and supported decision making when implementing the current Mental Health Act [8]. As the questions became more applied in nature, the level of knowledge became lower in terms of interventions that facilitate supported decision making in practice and even more significantly, in relation to experience with delivering or experiencing interventions that facilitate supported decision making in practice.

Interventions of similarity that arose from the hui themselves were categorised into the following five domains: (i) proactive pre-event planning/post-event debriefing; (ii) enabling options and choices; (iii) information provision; (iv) facilitating conditions and support to make a decision; and (v) education. 

### 3.1. Proactive Pre-Event Planning/Post-Event Debriefing

The similar attributes of this category of interventions included the need for any such processes to be widespread (‘*routine for anyone involved in care’*) and perhaps supported by a major campaign in the first instance. Whilst advance directives were specifically identified, participants also identified that various formats should be made available to cater for service-user preferences (e.g., ‘*a pre-recorded video of your ‘well’*
*self giving your ‘unwell’ self encouragement, advice and insight’; ‘an interactive PDF’)*. Service users needed to be provided with choice as to who to involve in such planning (e.g., family, whānau and significant others—‘*done with whānau selected by service user’*) and service providers needed to be responsive if that included them (‘*can be facilitated with GP, whānau, key worker in community team*’). The time and places for when and where this planning could occur and be reviewed needed to be flexible (‘*Post-intervention debrief—popping round for a chat and home visit after a significant event*’).

Peer support workers and whānau ora (family health) practitioners were identified as those who could potentially be responsible for liaising with service users for the purpose of facilitating the process of initiating and organising this type of planning and a regular review process. It was considered essential that there was some reliable way for all providers to be alerted, and have access to, any such plan (‘*human rights respect bracelet (protection) (e.g., like the diabetic medic alert bracelet)/card in wallet*’). Another similar attribute of importance to stakeholders was that any deviation from wishes expressed in a plan be fully justified and recorded (‘*where an advanced directive is not met—there is a documented process of attempts made at every stage*’). 

### 3.2. Enabling Options and Choices

The similar attributes of this category of interventions included highlighting that choice could only be meaningfully facilitated if a wide range of options were available; and services were re-configured to be option enabled and focused (‘*revamp paradigm to be option focused*’). This included options in relation to access, place of service and interventions (‘*having more resource and options after hours*’). In terms of interventions, both more non-pharmaceutical (e.g., ‘*culturally appropriate therapeutic interventions*’) and pharmaceutical options were considered necessary. Particular areas in need of more available options included acute services and settings (e.g., ‘*ability to stay safely at home*’); cultural- and spiritual-based supports (e.g., ‘*your marae*’—places of encounter and engagement; sites of central importance to Māori to express cultural and spiritual ways of achieving peace, unity and celebration), and peer support availability across service settings (‘*peer support available as part of all services—acute, GP etc.*’). 

### 3.3. Information Provision

Full and non-biased information about rights, access and treatment options were key similar attributes of this category of interventions (‘*develop standard info resources on types of interventions/options available +/or suitable for you*’). Examples of resources in support of such information being presented in a manner that facilitated clarity and understanding included pathways, maps and menus; and information being available in various mixed media (e.g., videos and cartoons) (‘*Need to visually see what support is available in your community*’). These resources and the time and support to comprehend them were considered essential to enabling fully informed supported decision making (‘*time and resources to support information to be conveyed in support of choice*’; ‘*sometimes we don’t know what we need (what supports) so help by showing us we can make a decision*’). 

### 3.4. Facilitating Conditions and Support to Make a Decision

The strong predominant similar attribute of this category of interventions was the provision of time to support the exploration of options and the making of decisions (‘*change time pressure/slow it down*’; ‘*delay in MHA* [Mental Health Act] *process*’; ‘*time to make decision*’; ‘*period of time for service user to think*’; ‘*when they are ready—no cut off*’; ‘*no deadlines = no pressure*’; ‘*providing time to explore all a person’s options before they have to make a decision*’). Other similar attributes included the need for the right space to be facilitative of decision making. Space was defined as places that provided for emotional, physical and spiritual safety (e.g., a ‘*crisis café*’; a ‘*quiet nice space, option of marae*’; ‘*where—right place, needs to be flexible, anywhere that’s right*’. Support to make decisions was considered as needing to be relationship/partnership (as opposed to power) focused (‘*person to person power balance*’; ‘*work alongside—journey together*’; ‘*therapeutic rapport—about relationships not power*’) and inclusive of service and support providers, including peer support workers particularly, and family, whānau and significant others (‘*crisis availability of peer support person*’; *include all without one it will not work*’). The opportunity, and time available being facilitative of the opportunity, to fully engage with people to establish the necessary understanding and trusting relationships that would then enable and support the exploring of options, such as the weighing up of pros and cons, was deemed important (‘*allowing time to build a trusting relationship between person and their clinician*’; ‘*service provider is open-minded and gets to know the person—their wants, dreams*’). Having regular chances to reflect on the outcomes of decisions made and the flexibility to change decisions were also similar attributes of this category of interventions (‘*providing time to reflect on decision outcomes and flexibility to make ongoing changes*’). 

### 3.5. Education

The similar attribute of this category of intervention was that all stakeholders (service users, clinicians, family, and whānau) were considered to be in need of general education about supported decision making, rights and legislation, and treatment options (e.g., ‘*clinical staff on processes for supporting informed consent*’; ‘*reduction of stigma*’). 

Table 3 shows the weighting rank for each category of interventions by perspective and total.

The category of facilitating conditions and support to make a decision was the most prioritised overall and by stakeholders from seven of the overarching perspective categories (service user, whānau, female, service provider, youth, consumer worker, and academia). The next most prioritised categories were proactive pre-event planning/post-event debriefing (Māori, family, Pasifika, and consumer workers) and information provision (Māori and LGBTQIA+). The categories of enabling options and choices and education were the least prioritised, with both of these not being identified as a most preferred option by any of the overarching perspective categories. When the overarching category of service providers was further broken down, the two differing results from that overall were that psychiatrists prioritised the category of proactive pre-event planning/post-event debriefing and nurses the category of enabling options and choices.

## 4. Discussion

In response to states or regional organisations that have ratified the CRPD being obliged under international law to abolish substitute decision making and replace it with supported decision making, and the dearth of research and literature focused on informing how that may best be facilitated, the present study involved a co-developed, exploratory, mixed-methods design that supported a bottom-up process, whereby key stakeholders came together to identify and then prioritise interventions facilitative of supported decision making. This work is particularly important in the Aotearoa New Zealand context presently as the process of repeal and replacement of the Mental Health Act is occurring, with strong indications that supported decision making will be a key feature of the new legislation. It is hoped that the findings of the present study will support the operationalisation of the legislative change in practice. 

The categories of intervention identified were proactive pre-event planning/post-event debriefing, enabling options and choices, information provision, facilitating conditions and support to make a decision; and education. Apart from proactive pre-event planning/post-event debriefing, all the other categories are more reflective of overarching processes facilitative of supported decision making as opposed to discrete interventions, although the similar attributes associated with the category of proactive pre-event planning/post-event debriefing are actually also very process oriented. These processes could in fact, and perhaps most appropriately, be facilitated by a variety of different interventions. This is consistent with all people being different and requiring different supports as is reflected by Article 12, paragraph 3, of the CRPD not specifying what form support should take and the committee of the CRPD guidance being that support is a broad term that encompasses both informal and formal support arrangements, of varying types and intensity [5]. 

Facilitating conditions and support to make a decision was the category most prioritised overall and by stakeholders from seven of the overarching perspective categories. The overwhelming key attribute associated with this category was the need for time in support of making a decision. Time seems like such an abstract concept and yet arguably it is a key attribute that most people would consider as necessary to support the making of major life decisions generally. Perhaps the reason it is prioritised here is because, relative to all the other categories of intervention and attributes of those, it is the attribute that is most commonly and consistently felt to be denied, especially in crisis situations. The presumption underpinning this is that these situations are of such a nature that they do not allow for the provision of time as a condition in support of decision making. However, this is not necessarily a reflection of service-user need. It does reflect the clinical reality of provision of service usually being stretched; and traditional decision makers, often medical staff, constantly triaging work to available time. In other work, psychiatric trainees have identified the lack of time as being a key barrier to reducing the use of coercive practices and applying supported decision making in practice [31]. This, and the social mechanisms that enable more rapid decisions to be made (such as mental health legislation) can easily lead to an apparently justified restriction in time to make such choices. However, the examples of crisis services that have been identified as promoting person-centred and rights-based approaches to mental health systems included in the recently published World Health Organisation resource reflect not only the lack of the need for the use of compulsory treatment in these situations but proactive approaches being taken to reduce the use of medication generally [22].

Human rights have become divided into those that are civil and political (including such matters as the right to liberty) and those that are economic, social and cultural (such as rights to housing), the distinction being that rights in the former group cannot be defeated by any lack of resources. Accordingly, Article 4(2) of the CRPD indicates that rights which are economic, social and cultural are to be implemented by the state using ”the maximum of its available resources” to realise them progressively, whereas civil and political rights, such as the right to equal recognition before the law (which requires substitute decision making being abolished and replaced with supported decision making), have no such resource-based limitations [3].

The fact that, when the overarching perspective category of service providers is broken down, psychiatrists identify proactive pre-event planning/post-event debriefing as their top preference likely reflects that this will be the most time efficient way for them to ascertain the will and preferences of service users in order that they can then be facilitative of supported decision making. However, the use of these types of interventions will never be sufficient to be fully facilitative of supported decision making and, hence the fact that clinicians being time poor is compromising their ability to respect the human rights of patients will need to be addressed in some other ways also. 

It is perhaps obvious, but supported decision making cannot occur unless options and choices are actually available and hence the enabling options and choices category. The similar attributes of that category highlight that such options and choices, while inclusive of those related to the bio or the biopsychosocial models of health, also need to extend beyond that, and more specifically to culturally- and spiritually-based supports. Whilst a consistently identified intervention, this category was not prioritised by any overarching stakeholder group but it was prioritised by nurses within the overarching category of service providers. This perhaps reflects that nurses are a group whose role makes them acutely aware of what options and choices are actually available and that this is an area that they experience in their work as lacking.

Similar attributes across categories could be summarised as the importance of easily and variably accessible options and choices (e.g., more acute service and settings) that involve the right people (including service and support providers, peer support workers especially, family, whānau and significant others), and the right relationships with (e.g., partnerships that involve understanding and trust), and the appropriate responsiveness of, those people; at the right place/space (e.g., those that provide for emotional, physical and spiritual safety); at the right time and for enough time; and the right material resources (e.g., information that is full, non-biased and presented in mixed media). The opportunity for regular review of, and reflection on, communicated and applied decisions was another attribute identified as important across multiple categories.

These results are consistent with a number of the rights-based critical elements proposed for response to mental health crisis based on crisis response practices that research shows—or that clinical and advocacy practice suggests—are anchored in human rights [18], including particularly communication and dialogue, presence (“being with”), flexible location, safe spaces of respite, and peer involvement.

The final category of education generally was consistently identified but not prioritised by any. Arguably this suggests that stakeholders are aware that, whilst important, education on its own is not a sufficient intervention to be facilitative of supportive decision making.

In Aotearoa New Zealand, interlinked health, social, and economic inequities and discrimination are disproportionally experienced by people from Māori, Pasifika, and LGBTQIA+ perspectives. It is of note that all three categories of stakeholders from these perspectives prioritised the intervention categories of proactive pre-event planning/post-event debriefing (Māori, Pasifika) and/or information provision (Māori, LGBTQIA+). This perhaps reflects that all or most of the categories are more elusively experienced by people from these perspectives; and that exclusion from decision making and marginalisation results in there being an enhanced need to be informed and to have a dedicated mechanism for input into the process to ensure it occurs, or in other words, to ensure the having of a voice. This highlights the importance for explorations of how supported decision making is to be facilitated needing to be inclusive, and responsive, to the various perspectives that stakeholders come from. Cultural, social and structural determinants of health and well-being are important and necessarily introduce an element of complexity that does not permit simplified solutions. It is acknowledged that these findings will not be generalisable to stakeholders from other jurisdictions and, in this sense, each country needs to plot their own way forward. 

Strengths of this study include the range of stakeholders who were involved (including particularly the significant number of service users), as reflected through both the demographic data and the data associated with the prioritisation of interventions. Ultimately, supported decision making will involve all such stakeholders being committed, and working together, to facilitate it. The significant number of participants from a Māori perspective is particularly important for a study being conducted in the Aotearoa New Zealand context. Of note is the lack of stakeholders attending specifically in the capacity of family or whānau and yet at least ten or more participants identified this as a perspective they came from through the data collection associated with the prioritisation of interventions. Age and time in the sector of participants reflect that most were older and with considerable sector experience. This is a strength in terms of that experience informing this study; however, the corresponding limitation is the lack of youth and people new to the sector having participated and informed the work. Gender was another area that lacked diversity, with significantly more females than males having participated; and little to no other identified genders. This may reflect the concept of supported decision making being one that females are more receptive to. A key stakeholder group that we did not target and for which there were very few participants (and not sufficient for inclusion as an overarching perspective category) were managers, funders and planners. Arguably, it is crucial that they too be engaged in this work in order to have the top-down support for the developments required to enable the facilitation of supported decision making in practice. This is consistent with committed senior management being identified as a decisive factor for success in Broberg et al.’s pilot study of applying a human rights-based approach to psychiatric care [20]. It is notable that the findings of this study are likely to come only with considerable resource allocation.

As well as the methodology supporting multiple stakeholder involvement in the hui, this aspect of the work was also enabled through the co-produced approach, whereby the researcher team came from different perspectives, and those different perspectives informed and supported all aspects of the research process. A further strength was the bottom-up and exploratory nature of the process, resulting in their being no limitations based on current practice, and addressing this gap in the current literature. Participants being able to self-identify the perspectives they considered themselves as coming from was a strength; however, the semi-arbitrary approach of then grouping them as a result of this self-definition is a limitation. The use of hui as the procedure for knowledge generation was a strength, particularly in terms of how it serves to support the participation of those whotend to be marginalised through conventional spaces and methods of engagement. However, some stakeholders from particular perspectives were not well represented and it would be prudent to consider some further targeted exploration of those groups. In addition, feedback of participants from Māori perspectives identified that they felt it would be valuable to have Māori-only hui to support a specific focus on their perspectives in relation to supported decision making. Dot voting (“dotmocracy”) has a number of limitations, although this was mitigated to a certain extent by use of an advanced approach; and independent coding by multiple persons to inform the thematic analysis was not undertaken, although this was somewhat mitigated through review of the lead researcher’s provisional outputs by the rest of the team. Hence, the methodology was considered satisfactory for the exploratory nature of this study.

Despite the lack of applied knowledge, and experience, in relation to delivering or experiencing interventions that facilitate supported decision making in practice pre-hui, the work that stakeholders produced through this study was substantive and reflective of them having a significant contribution to make in advancing this area of development. It bodes well for having a reasonable groundswell of support for the actual application of supported decision making in practice. However, participation was voluntary so these people are those who are receptive to, and interested in, this subject area already. Remaining aware that there will be others who are equally disinterested or even, actively opposed, is vitally important. Given this, a prudent way forward is to task those who are receptive and interested with leadership and advocacy roles in support of service-wide implementation. Other enablers will also be critical. Ultimately, the application of supported decision making in practice is going to have significant resource implications, particularly in terms of time, people, and space.

Action research examining the effectiveness of supported decision-making interventions is also going to be important. Authors of a recent paper considering appropriate methodologies for the researching of shared decision making, which admittedly is a different but similar concept to supported decision making, recommend that the re-thinking of medical trial designs is needed [32]. Consistent with the co-produced methodology employed in the present study, the involvement of service users and carers in such re-thinking and trial participants as co-investigators in studies, are both identified as being required. Furthermore, they recommend a number of conditions for what they refer to as ‘pragmatic’ trials, including allocating on the basis of choice (e.g., consider use of partially randomised preference trial designs), stop double-blinding and respect trial participants’ unblinded opinion (e.g., consider single-blinded designs with objective outcome measures, obtained by a blinded research team), use outcome measures that are relevant for trial participants (consult with service users in order to determine this), minimise the number of exclusion criteria, use statistics that reflect the clinical complexity (e.g., instrumental variable analyses can better accommodate non-dichotomous events), and give tailored and user-friendly information before inclusion. Of particular note is that the detailed specification of this last condition includes the offering of time to think and discuss participation with caregivers.

## 5. Conclusions

Through this exploratory study, service users and service providers, inclusive of those that also came from family and whānau perspectives, identified and then prioritised interventions considered facilitative of supported decision making. The categorisation of those identified—proactive pre-event planning/post-event debriefing, enabling options and choices, information provision, facilitating conditions and support to make a decision, and education—reflects a process focus. Overall, the majority of the different categories of stakeholders prioritised facilitating conditions and support to make a decision as having the potential to lead to the greatest improvements in the lives of people who experience mental distress and their loved ones. However, people from Māori, Pasifika, and LGBTQIA+ perspectives, who disproportionally experience inequities and discrimination, prioritised the intervention categories of proactive pre-event planning/post-event debriefing and/or information provision, which, in addition to the actual finding, highlights the critical need for exploration and application of supported decision making in practice being proactively inclusive of, and responsive to, different stakeholder perspectives and needs. Similar attributes across categories of intervention entailed the importance of easily and variably accessible options and choices that involve the right people, and the right relationships with, and the appropriate responsiveness of, those people; at the right place/space; at the right time and for enough time; with the support of the right material resources; and opportunities for regular reviews of, and reflection on, communicated and applied decisions. The need for time in support of making a decision was especially highlighted and yet this is currently impeded by clinical time being insufficient to allow for such. A lack of resources, including human resources, is not a valid defence for not implementing what is required to enable supported decision making in practice in accord with Article 12 of the CRPD. The application of supported decision making in practice is going to have significant resource implications, particularly in terms of time, people, and space. Another critical enabler is the leadership and advocacy of stakeholders, who are most receptive and interested, to support implementation. Taking an action research approach to the development and implementation of interventions based on what has been identified through this exploratory study, using re-thought research designs that are pragmatic, has the potential to make supported decision making a reality in Aotearoa New Zealand, and elsewhere. 

## Figures and Tables

**Table 1 ijerph-19-06002-t001:** Demographic characteristics of study participants.

Demographic	Number
**Age**	
18–24	<5
25–34	8
35–44	22
45–54	26
65 or above	17
**Gender**	
Female/Wāhine	55
Male/Tāne	26
Non-binary	<5
**Ethnicity** ^1^	
Pākeha (European of New Zealand descent)	46
Māori	19
Pasifika	6
Asian	8
Other migrant peoples (e.g., Middle East, Latin America, Africa)	<5
Other	13
Prefer not to say	<5
**Capacity attending in** ^2^	
Service provider ^3^	26
Consumer/peer/service user/lived experience/tangata whaiora	19
Consumer/cultural advisor	10
Peer support worker	6
Academic	6
**Amount of time in sector**	
Under 1 year	5
1–2 years	<5
3–5 years	13
5–10 years	12
11–15 years	8
16–20 years	13
Over 20 years	27

^1^ Participants are instructed to tick all that apply. ^2^ Categorised self-definitions, with categories of two or less not having been included. ^3^ Excludes peer support workers (included as separate category).

**Table 2 ijerph-19-06002-t002:** Participant responses to pre-hui survey about level of knowledge in respect of supported decision making and any interventions considered to be currently in existence.

	No Knowledge (I Have No Idea What This Means)	A Little Knowledge (I’ve Heard about This, But Couldn’t Explain it to You)	Average Knowledge (I’m Familiar with This and Have OK Working Knowledge about It)	Very Good Knowledge (I’m Pretty Familiar with This and Can Explain It to an Adequate Degree)	Expert Knowledge (I Know a Lot about This and It Regularly Informs Many Conversations and Activities in My Professional Role)	Total
Current level of knowledge of supported decision making	5	22	31	8	7	73
Current level of knowledge of the difference between substitute decision making and supported decision making	14	21	23	7	6	71
Current level of knowledge of the difference between legal capacity and mental capacity	9	18	29	10	6	72
Current level of knowledge of the key principles that underpin supported decision making	9	31	21	5	6	72
Current level of knowledge of the status of supported decision making in New Zealand	22	32	11	5	1	71
Current level of knowledge of the Human Rights and the Mental Health (Compulsory Assessment and Treatment) Act 1992 guidelines [8]	1	11	34	17	8	71
Current level of knowledge of interventions that facilitate supported decision making in practice	7	30	24	6	4	71
Current level of experience with delivering or experiencing interventions that facilitate supported decision making in practice	13	27	18	11	4	73

**Table 3 ijerph-19-06002-t003:** Weighting rank for each category of interventions by perspective and total.

	Intervention Categories				
Perspectives	Proactive Pre-Event Planning/ Post-Event Debriefing	Enabling Options and Choices	Information Provision	Facilitating Conditions and Support to Make a Decision	Education
Service user (includes participants who self-identified as service user, lived experience, patient, peer)	3	5	2	1	4
Māori	1	4	1	3	5
Family (includes participants who self-identified as family, parent, migrant parents, partner)	1	5	3	2	4
Whānau (includes participants who self-identified as whānau, whānau lived experience)	4	5	2	1	3
Female (includes participants who self-identified as female, wahine)	4	5	2	1	3
Service provider (includes participants who self-identified as nurse, psychiatrist, doctor, clinician, clinical, occupational therapist, service provider, therapist, counsellor, community mental health nurse, community GP, psychologist, community support worker, C/L nurse, social worker, whānau support worker	2	4	5	1	3
Psychiatrist	1	2	5	2	4
Doctor (includes participants who self-identified as doctor, community GP)	5	2	4	1	3
Clinician (includes participants who self-identified as clinician, clinical)	2	4	3	1	5
Nurse (Includes participants who self-identified as nurse, community mental health nurse, C/L nurse)	3	1	5	4	2
Therapist (includes participants who self-identified as counsellor, therapist, psychologist)	2	5	3	1	3
Other (includes participants who self-identified as service provider, occupational therapist, social worker, whanau support worker, community support worker	5	3	4	1	2
Youth	3	4	2	1	5
Pasifika	1	3	3	2	5
LGBTQIA+ (includes participants who self-identified as LGBTIQ, LGBTQIA)	5	2	1	3	3
Consumer worker (includes participants who self-identified as youth consumer advisor, peer support, consumer worker, LE work role)	1	5	3	1	4
Academia (includes participants who self-identified as academia, researcher, legal mental health research)	2	5	3	1	5
Summed score	27	47	27	17	47
Rank weight	2	4	2	1	4

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
