# Peer review of "From Substitute to Supported Decision Making: Practitioner, Community and Service-User Perspectives on Privileging Will and Preferences in Mental Health Care"

_ijerph, 2022, doi:10.3390/ijerph19106002_

Round 1
Reviewer 1 Report
Thank you for submitting your manuscript “Human Rights: Exploring the application of supported decision-making in practice” to the International Journal of Environmental Research and Public Health. I am glad to be given this opportunity to read and review this manuscript. This manuscript highlights an important issue regarding the ratification and implementation of the CRPD relating to interventions that adopt supported decision-making rather than substitute decision-making.
The manuscript is well-written and the issues such as word choice/order, subject-verb agreement, tense, consistency, clarity, sentence structure, etc. have been excellently addressed.
Author Response
That for this review. We appreciate it.
Kind Regards
Sarah
Reviewer 2 Report
Useful paper and important topic.
Some comments: The title is terribly vague and should be revised to indicate what the paper is actually about (as closely as possible so that researchers that want to know about this or related issues can find it).
Did you use any guiding approach/semi-structured questionnaire for the data collection?
Regarding the analysis, I would expect to see more information about the coding strategy employed. Did you use two cycles? What was the approach taken in each? Consult literature on qualitative analysis as needed - the focus should be on providing as much information as possible so that the approach and results are replicable. Speaking of which - would the data corpus and analysis materials be made available to scholars on request, or is this prohibited (it should be mentioned in the text).
Author Response
- Useful paper and important topic. Response: Thank you
Some comments: The title is terribly vague and should be revised to indicate what the paper is actually about (as closely as possible so that researchers that want to know about this or related issues can find it). Response: The title has been revised.
Did you use any guiding approach/semi-structured questionnaire for the data collection? Response: More detail has been added to the methods section in response to this query
Regarding the analysis, I would expect to see more information about the coding strategy employed. Did you use two cycles? What was the approach taken in each? Response: More detail has been added to the methods in response to this comment.
Consult literature on qualitative analysis as needed - the focus should be on providing as much information as possible so that the approach and results are replicable. Speaking of which - would the data corpus and analysis materials be made available to scholars on request, or is this prohibited (it should be mentioned in the text). Response: Our informed consent did not include getting consent from participants for the materials being used in this way so I don't think we would be able to do this.
Reviewer 3 Report
Thank you for the opportunity to review this paper on support decision making for people with mental illness. The paper is very well written. The study uses a novel method which is culturally appropriate for Aotearoa New Zealand. I have a few suggestions.
The Introduction section contains a lot of lists of factors/variables/elements identified in the literature as relevant to the topic. Rather than listing these, can you re-write some of this section to distil the critical points? The lists make for laborious reading.
The Method section requires some additional information about how the information during the hui's was recorded for analysis. It is unclear whether the discussions were recorded and transcribed or whether some other method was used. The same is required for the collection of the quantitative data.
The Results section contains very limited qualitative data to demonstrate each of the five domains. Indeed only domain IV contains any actual qualitative data. I would suggest adding some other examples for the other domains.
The Discussion section is excellent.
Author Response
Thank you for the opportunity to review this paper on support decision making for people with mental illness. The paper is very well written. The study uses a novel method which is culturally appropriate for Aotearoa New Zealand. I have a few suggestions. Response: Thanks for this feedback.
The Introduction section contains a lot of lists of factors/variables/elements identified in the literature as relevant to the topic. Rather than listing these, can you re-write some of this section to distil the critical points? The lists make for laborious reading. Response: Thank you for this feedback. Some of the lists have been removed and replaced with the critical points having been distilled. Some of the lists remain as difficult to convey the material in a different manner however this section has been re-written to try and address the issue of the laborious reading.
The Method section requires some additional information about how the information during the hui's was recorded for analysis. It is unclear whether the discussions were recorded and transcribed or whether some other method was used. The same is required for the collection of the quantitative data. Response: The additional information identified as being required has been added to the methods section.
The Results section contains very limited qualitative data to demonstrate each of the five domains. Indeed only domain IV contains any actual qualitative data. I would suggest adding some other examples for the other domains. Response: Other examples have been added to all the other domains.
The Discussion section is excellent. Thank you.
Reviewer 4 Report
Thank you for the opportunity to review this important piece of research examining what may facilitate operationalisation of supported-decision making within mental health systems. Within this, it was noteworthy that the research was co-designed and culturally informed by groups recognised as being more likely to experience poorer determinants of mental health that places them at greater risk of experiencing mental distress.
As a reviewer not familiar with the current mental health act and legislation of NZ, I would have found it useful for the paper to provide a brief overview within the introduction of the status of the act and proposed changes to the legislation including alignment with Article 12. This could then be used to position the research in terms of how it might inform these processes, including making mention of this within the discussion. Whilst I appreciated the broad application of the research beyond NZ, I still feel a paragraph on any facilitators/barriers to the operationalisation of the findings in both informing policy/legislation and practice in the NZ context would also be useful.
On page two when you introduce the literature to date, I felt that it might be more useful to start with the lit review findings, before talking about the pilot study. More details of the context of the pilot study would also be appreciated.
The WHO community based mental health resource was mentioned, but I wondered also if the researchers draw any lessons or potential usefulness from the WHO Quality Rights Toolkit for progressing policy/legislative and operational changes in the context of the research? This doesnt have to inform changes within the current manuscripit, but I'd be interested to know.
In terms of the findings, I wondered whether it would be useful to present the five domains in the order they were prioritised?
The second sentence of the second paragraph of the discussion was clearly relevant but quite confusing to try and interpret. I wonder if it could be unpacked more clearly across two sentences?
Author Response
Thank you for the opportunity to review this important piece of research examining what may facilitate operationalisation of supported-decision making within mental health systems. Within this, it was noteworthy that the research was co-designed and culturally informed by groups recognised as being more likely to experience poorer determinants of mental health that places them at greater risk of experiencing mental distress. Response: Thank you for this feedback
As a reviewer not familiar with the current mental health act and legislation of NZ, I would have found it useful for the paper to provide a brief overview within the introduction of the status of the act and proposed changes to the legislation including alignment with Article 12. This could then be used to position the research in terms of how it might inform these processes, including making mention of this within the discussion. Whilst I appreciated the broad application of the research beyond NZ, I still feel a paragraph on any facilitators/barriers to the operationalisation of the findings in both informing policy/legislation and practice in the NZ context would also be useful. Response: More detail about the NZ legislation has been added to both the introduction and the discussion in response to this comment.
On page two when you introduce the literature to date, I felt that it might be more useful to start with the lit review findings, before talking about the pilot study. More details of the context of the pilot study would also be appreciated. Response: This section has been completely re-written in response to this feedback and that of another reviewer.
The WHO community based mental health resource was mentioned, but I wondered also if the researchers draw any lessons or potential usefulness from the WHO Quality Rights Toolkit for progressing policy/legislative and operational changes in the context of the research? This doesnt have to inform changes within the current manuscripit, but I'd be interested to know. Response: This has been referred to in the introduction.
In terms of the findings, I wondered whether it would be useful to present the five domains in the order they were prioritised? Response: We would prefer to leave it as is given our commitment to stakeholders from minority perspectives and the fact that there was some variability of prioritisation across groups
The second sentence of the second paragraph of the discussion was clearly relevant but quite confusing to try and interpret. I wonder if it could be unpacked more clearly across two sentences? Response: This has been addressed as suggested